

# Utility of Geostationary Lightning Mapper Derived Lightning NOx Emission Estimates in Air Quality Modeling Studies

Peiyang Cheng[1,2], Arastoo Pour-Biazar[3], Yuling Wu[3], Shi Kuang[3], Richard T. McNider[2], William J. Koshak[4]

[1]Zhejiang Climate Center, Zhejiang Meteorological Bureau, Hangzhou, Zhejiang, 310052, China
[2]Department of Atmospheric and Earth Science, University of Alabama in Huntsville, Huntsville, AL, 35805, USA
[3]Earth System Science Center, University of Alabama in Huntsville, Huntsville, AL, 35805, USA
[4]Earth Science Branch, NANA Marshall Space Flight Center, Huntsville, AL, 35808, USA

*Correspondence to*: Peiyang Cheng (peiyang.cheng@hotmail.com; peiyang.cheng@nsstc.uah.edu)

**Abstract.** Lightning is one of the primary natural sources of nitrogen monoxide (NO), and the influence of lightning-induced NO (LNO) emission on air quality has been investigated in the past few decades. In the current study an LNO emissions model, which derives LNO emission estimates from satellite-observed lightning optical energy, is introduced. The estimated LNO emission is employed in an air quality modeling system to investigate the potential influence of LNO on tropospheric ozone. Results show that lightning produces 0.174 Tg N of nitrogen oxides ($NO_x$ = NO + $NO_2$) over the contiguous U.S. (CONUS) domain between June and September 2019, which accounts for 11.4% of the total $NO_x$ emission. On average, $LNO_x$ emission increases tropospheric ozone concentration by 1–2% (or 0.3–1.5 ppb) in the column; the enhancement is maximum at ~4 km above ground level, with a minimum near the surface. The southeast U.S. has the most significant ground-level ozone increase, with up to 1 ppb (or 2% of the mean observed value) difference for the maximum daily 8-hour average (MDA8) ozone. However, many of these numbers are near the lower bound of the uncertainty range given in previous studies, suggesting the current LNO production rate used in the LNO emissions model may need to be adjusted. Moreover, the episodic impact of LNO on tropospheric ozone can be considerable. Performing backward trajectory analyses revealed two main reasons for the significant ozone increase: long-distance chemical transport and lightning activity in the upwind direction shortly before the event. In addition, the mixing of high $LNO_x$ (or ozone) plumes is likely another reason for ozone enhancement.

## 1 Introduction

The air quality community is concerned about nitrogen oxides ($NO_x$), a group of highly reactive gases – nitric oxide (NO) and nitrogen dioxide ($NO_2$) – with $NO_2$ being regulated by the U.S. Environmental Protection Agency (EPA) as one of the criteria air pollutants. One reason is that, in the presence of sunlight and water vapor, $NO_x$ can react with volatile organic compounds (VOCs) to produce ozone ($O_3$), a secondary air pollutant that has adverse health effects on susceptible individuals (Chen et al., 2007; Post et al., 2012; Caiazzo et al., 2013; EPA, 2015, 2021) and is harmful to the environment



(Van Dingenen et al., 2009; Fuhrer et al., 2016; Dinan et al., 2021). After decades of efforts to reduce anthropogenic $NO_x$ emissions in the U.S. (Simon et al., 2015), the relative importance of naturally emitted $NO_x$ to air quality is expected to increase (Kang et al., 2019a). Lightning, an electrical discharge phenomenon caused by charge separation and accumulation during a thunderstorm (Verma et al., 2021), is an important natural source of $NO_x$ (Pour-Biazar and McNider, 1995). The

intense heating and subsequent rapid cooling of air that occur due to a lightning discharge convert stable nitrogen ($N_2$) and oxygen ($O_2$) into NO (Bond et al., 2001). It was estimated that lightning-induced $NO_x$ ($LNO_x$) emission accounts for 10-15% of the global $NO_x$ budget (Schumann and Huntrieser, 2007) and more than 80% of the upper-tropospheric $NO_x$ in summer (Cooper et al., 2009).

Because $NO_x$ production from lightning is sensitive to various factors, such as peak current, channel length, strokes per flash,

air density, and energy dissipation rate (Cooper et al., 2009; Koshak et al., 2014a, 2015; Murray, 2016), the amount of $NO_x$ produced from lightning is still highly uncertain, even though considerable research efforts have been devoted to quantifying the amount of $NO_x$ produced by lightning flashes, including theoretical calculations (e.g., Chameides et al., 1977), laboratory experiments (e.g., Peyrous and Lapeyre, 1982), cloud-scale chemical transport model simulations (e.g., Ott et al., 2010), ground-based observations (e.g., Wada et al., 2019), and satellite-based columnar measurements (e.g., Bucsela et al., 2010;

Pickering et al., 2016). A comprehensive literature review by Schumann and Huntrieser (2007) reported that the best estimate of the $LNO_x$ production rate is $15 \times 10^{25}$ molecules of $NO_x$ per flash with uncertainty factors ranging from 0.13 to 2.7, which is equivalent to 250 (32.5–675) moles of $NO_x$ production per flash. A subsequent study by Murray (2016) updated the uncertainty range to be 17–700 moles of $NO_x$ production per flash.

With the availability of lightning flash data from the National Lightning Detection Network (NLDN) (Orville et al., 2002,

2011), a reputable ground-based lightning detection network that has a high (~90–95%) cloud-to-ground (CG) flash detection efficiency (DE) over the contiguous U.S. (CONUS), various models and schemes have been developed to estimate $LNO_x$ emission and investigate its impact on ozone prediction in regional chemical transport models (Kaynak et al., 2008; Smith and Mueller, 2010; Allen et al., 2012; Koshak et al., 2014a; Wang et al., 2015; Kang and Pickering, 2018; Kang et al., 2019a,b, 2020). For instance, Allen et al. (2012) introduced an $LNO_x$ parameterization scheme, which utilizes monthly

NLDN (mNLDN) flash data, into the Community Multiscale Air Quality (CMAQ) model (Byun and Schere, 2006; Appel et al., 2021). The mNLDN scheme assumes that total column $LNO_x$ emission is proportional to model-predicted convective precipitation (CP) with local adjustment so that the monthly average CP-based flash rate in each model grid cell matches the NLDN-based monthly mean total flash rate. The total column $LNO_x$ emission is then distributed vertically based on a preliminary version of the segment altitude distributions (SADs) derived by Koshak et al. (2014a) using North Alabama

Lightning Mapping Array (NALMA) data (Goodman et al., 2005). Kang et al. (2019a) simplified the mNLDN scheme in CMAQ by using only gridded hourly NLDN (hNLDN) flash data to ingest $LNO_x$ emission into model grid cells directly. However, since the hNLDN scheme is not dependent on the model-predicted CP field, discrepancies between the time and location of the released $LNO_x$ emission and convective activity, as well as other convectively transported ozone precursors, may exist. In addition, Kang et al. (2019a) also introduced a parameter scheme (pNLDN) that is based on linear and log-



linear regression parameters derived from multiyear NLDN lightning flash data and the model-predicted CP field, which can
be used when lightning observations are not available (such as air quality forecasts and future climate studies).

Further, satellite-based lightning observations can also be used to estimate $LNO_x$ production (e.g., Bucsela et al., 2010;
Pickering et al., 2016; Koshak et al., 2014b; Koshak, 2017). Koshak et al. (2014b) and Koshak (2017) proposed an approach
that derives $LNO_x$ emission estimates independent of model fields using satellite lightning imager flash optical energy data

[e.g., as from the Tropical Rainfall Measuring Mission (TRMM) Lightning Imaging Sensor (LIS; Cecil et al., 2014), and
geostationary lightning mappers (see below), respectively].  It is referred to as the β-method since it relies on computing a
scalar denoted as β that compensates for some of the uncertainties and converts the satellite-detected flash optical energy
(typically hundreds of femtojoules as measured from geostationary platforms) to an estimate of the total lightning flash
energy (typically gigajoules), and consequently to $LNO_x$. However, because TRMM/LIS is a low-Earth-orbiting satellite, it

cannot record the entire life cycle of a thunderstorm (Bucsela et al., 2010). As a result, using TRMM/LIS data cannot
explicitly characterize the diurnal variation of $LNO_x$ emission over a specific region. This limitation can be overcome by
using observations from Geostationary Operational Environmental Satellite R-series (GOES-R) Geostationary Lightning
Mapper (GLM), which has a similar instrument design and data processing algorithm to TRMM/LIS (Goodman et al., 2013;
Schmit et al., 2017). GLM is the first operational lightning mapper in the geostationary orbit and continuously monitors

lightning activity over the Americas and adjacent ocean regions. It collects lightning optical pulses at 777.4 nm (i.e., the
center of a prominent oxygen emission triplet in the lightning spectra) with a nadir staring, high-speed Charge Coupled
Device (CCD) array.

Our recent paper, Wu et al. (2023), introduced an offline $LNO_x$ emission model that utilizes GOES-16 and GOES-17 GLM
(hereinafter referred to as GLM-16 and GLM-17) lightning observations to prepare $LNO_x$ emission input for regional air

quality modeling systems by implementing the β-method introduced in Koshak et al. (2014b) and Koshak (2017). As a
follow-up study, this paper applies the GLM-estimated $LNO_x$ emission in air quality model simulations to study how it
would affect ozone simulation. One caveat is that the $LNO_x$ emission model does not constrain the time and location of
$LNO_x$ production using the model cloud field. Therefore, desynchronization between model clouds and $LNO_x$ emission
would likely introduce some uncertainty. This issue will be addressed in our future study by assimilating GOES cloud

observations (White et al., 2018, 2022) to improve model cloud placement. The rest of this paper is organized as follows:
Section 2 provides descriptions of the Wu et al. (2023) GLM-based $LNO_x$ emission model, Section 3 states how air quality
simulations are conducted, Section 4 presents simulation results and discusses the potential impact of $LNO_x$ emission on
ozone prediction, and Section 5 summarizes the key findings and lists future work.





## 2 GLM-based LNO$_x$ emission model

### 2.1 Column total LNO$_x$ production

The LNO$_x$ emission model described in Wu et al. (2023) first estimates column total LNO$_x$ production from GLM Level 2 data product, which is currently distributed at National Oceanic and Atmospheric Administration (NOAA) Comprehensive Large Array-data Stewardship System (CLASS) (https://www.class.noaa.gov, accessed 28 August 2022). GLM Level 2 data product contains the time, geographic location, areal coverage, and radiant energy information of three lightning elements –

event (pixel-level lightning registered by GLM over a 2-ms integration window), group (one or more simultaneous events detected in adjacent pixels), and flash (a set of sequential groups occurring within 330 ms and 16.5 km). Since the temporal resolution of the product is much higher than needed by air quality modeling systems, only flash-level data are processed to improve computational efficiency. Previous assessments (Marchand et al., 2019; Bateman and Mach, 2020; Bateman et al., 2021; Blakeslee et al., 2020; Murphy and Said, 2020; Zhang and Cummins, 2020; Rutledge et al., 2020) estimated that GLM

flash-level DE is greater than 70%, with better performance at night than during the day. However, a few studies (e.g., Murphy and Said, 2020; Bateman and Mach, 2020; Blakeslee et al., 2020) pointed out that GLM flash DE is significantly depleted on the edge of the sensor FOV (e.g., over the northwestern U.S. for GLM-16). The recent study by Wu et al. (2023) showed that significantly more (fewer) NLDN-detected CG flashes could be matched to GLM-16 flashes than GLM-17 flashes east (west) of 106.2°W. Therefore, to reduce the uncertainty caused by diminished GLM flash DE, GLM-16 flashes

east of 106.2°W and GLM-17 flashes west of 106.2°W are selected, merged, aggregated into hourly values, and gridded onto pre-defined model grid cells before subsequent calculations.

With the core assumption that the GLM-detected flash optical energy is proportional to the total flash energy (i.e., the total stored electrostatic flash energy typically measured in gigajoules, and that is released as acoustical and electromagnetic energy in the discharge), the amount of NO$_x$ (in moles) produced by flash $k$ ($P_k$) is estimated by Koshak et al. (2014b) and

Koshak, (2017) as

$$P_k = \frac{Y}{\beta_k N_A} Q_k \ , \tag{1}$$

where $\beta_k$ is the fraction of the total lightning-released optical energy detected by GLM for flash $k$, $N_A$ ($6.022 \times 10^{23}$ molecules per mole) is the Avogadro's number, $Y$ ($\sim 10^{17}$ molecules per Joule) is the thermochemical yield of NO$_x$ (Borucki and Chameides, 1984), and $Q_k$ is the GLM-detected optical energy (in Joules) from flash $k$ (provided by GLM flash optical

energy data). The only variable needed for obtaining the value of $P_k$ is the dimensionless scaling factor $\beta_k$, which is sensitive to various lightning and cloud scattering properties and GLM sensor characteristics (Koshak et al., 2014b; Wu et al., 2023). To make this method feasible, it is assumed that many (but not all) of these factors average out for a large number of GLM flashes and numerous types of thundercloud structures over diverse geographical areas. Assuming that the particular $\beta_k$ in Eq. (1) can be replaced by a fixed (mean) value $\beta$, then Eq. (1) can be re-written as



$P_k = \frac{Y}{\beta N_A} Q_k$ .                                                            (2)

It is important to note that this equation provides a variable flash-to-flash estimate of LNO$_x$ production (hence the k subscript in the production variable $P_k$). Only the value β is chosen as fixed. Now, to obtain a representative value of β, multiple years of GLM flash optical energy data are needed, and Eq. (2) is rewritten as

$\beta = \frac{Y}{N_A} \frac{\sum_{k=1}^{N} Q_k}{\sum_{k=1}^{N} P_k} = \frac{Y}{N_A} \frac{\sum_{k=1}^{N} Q_k}{N\bar{P}}$ ,                                              (3)

where $N$ is the total number of GLM flashes within an extended period (over the entire observational domain), and $\bar{P}$ is the average amount of NO$_x$ produced by lightning flashes. In recent air quality modeling studies, 250 to 500 moles per flash is typically used for $\bar{P}$ (Allen et al., 2010; Ott et al., 2010; Koshak et al., 2014b; Koshak, 2017; Wang et al., 2015; Zhu et al., 2019; Kang et al., 2019a,b, 2020). For this study, the LNO$_x$ emission model assumes that a lightning flash would produce 250 moles of NO$_x$ on average, a commonly-cited LNO$_x$ production rate in the literature (Schumann and Huntrieser, 2007).

Processing almost three years (February 2019 – December 2021) of GLM data within the CONUS yields an estimate of $1.53359 \times 10^{-22}$ for β. Once β is known, NO$_x$ production by each lightning flash is estimated using Eq. (2), and column total LNO$_x$ emission can be determined.

The derivation of the column total LNO$_x$ production has several sources of uncertainty. First, the fixed value of β is clearly biased by the mean LNOx production per flash [i.e., the value $\bar{P} = 250$ moles per flash assumed is highly uncertain

(Schumann and Huntrieser, 2007; Murray, 2016)]. Many variables can affect the value of $\bar{P}$. For example, NO$_x$ production is sensitive to various lightning characteristics, such as peak current, channel length, strokes per flash, air density, and energy dissipation rate (Cooper et al., 2009; Koshak et al., 2014a, 2015; Murray, 2016). A lightning discharge with a longer channel length or a higher peak current produces more NO$_x$. In the latter case, a higher peak current normally implies more area under the stroke current waveform $i(t)$ and therefore more net energy in the discharge; and this is explicitly true for the return

stroke current models for $i(t)$. Many studies have shown that one CG flash might produce up to 10 times more NO$_x$ than an intra-cloud (IC) flash (Koshak et al., 2014a; Carey et al., 2016; Lapierre et al., 2020) as CG flashes typically have stronger peak currents, longer channel lengths, more channel at lower altitude where the thermochemical yield is larger, and extend to a larger area than IC flashes (Rakov and Uman, 2003; Koshak et al., 2009, 2014a; Koshak, 2010; Mecikalski and Carey, 2018). So even though the β method nicely computes variable LNO$_x$ production on a per flash basis based on flash-specific

GLM-observed flash optical energy, still the value of $\bar{P}$ is biased/assumed and therefore contains uncertainty. Second, not all lightning flashes are detected by GLM. Recent studies indicated that GLM flash DE is correlated with the type, geometric size, optical energy, duration of the flash, cloud optical depth, seasons, time of day, and sensor viewing geometry (e.g., Blakeslee et al., 2020; Murphy and Said, 2020; Rutledge et al., 2020; Zhang and Cummins, 2020). For example, less energetic and shorter IC flashes are less likely to be detected than CG flashes. As a result, NO$_x$ production from any missed

flashes would not be counted. Third, since β is an average based on multi-year GLM flash optical energy data, it can be further refined as more GLM data become available. Despite all of the factors mentioned in this paragraph, an advantage of



the β-method is that all these uncertainties are accounted for by a single scalar (β), allowing potential improvements in future studies.

## 2.2 Vertical distribution of LNO$_x$ emission

Generally, air quality modeling systems require three-dimensional gridded emissions as input. Since GLM lightning observations are only two-dimensional, extra information is needed to distribute the derived column total LNO$_x$ emission vertically. This is accomplished by adapting monthly LNO$_x$ production profiles created by the Lightning Nitrogen Oxides Model (LNOM) (Koshak, 2010; Koshak et al., 2009, 2014a). The LNOM is a flash-based model that fuses laboratory results (Wang et al., 1998), theoretical results (Cooray et al., 2009), and additional simplifying assumptions discussed in Koshak et

al. (2014a) with Lightning Mapping Array (LMA) (Goodman et al., 2005) and NLDN lightning observations. In this study, pre-generated monthly LNOM profiles for CG and IC flashes archived at the National Aeronautics and Space Administration (NASA) Global Hydrology Resource Center (GHRC) (https://ghrc.nsstc.nasa.gov/uso/ds_docs/lnom/lnom_dataset.html, accessed 04 August 2022) are used to vertically distribute LNO$_x$ emissions. Meanwhile, to account for the different contributions of CG and IC flashes to NO$_x$ production, the climatological geographic distribution of daily IC-to-CG ratio

(denoted by the Z ratio) developed by Boccippio et al. (2001) with updates from Medici et al. (2017) is applied in conjunction with the LNOM profiles. Note that using the archived LNOM profiles and the climatological Z ratio introduces another layer of uncertainty to the derived three-dimensional LNO$_x$ emission. The LNOM profiles were constructed around the NALMA and therefore are more representative of Northern Alabama than other regions of the CONUS. Meanwhile, the Z ratio map was generated using multi-year satellite and ground-based lightning observations, but lightning activity varies

appreciably from year to year.

## 3 WRF-SMOKE-CMAQ model configuration

Air quality simulations were conducted by the modeling system containing the Weather Research and Forecasting (WRF) (Skamarock et al., 2021), the Sparse Matrix Operator Kernel Emissions (SMOKE) (https://www.cmascenter.org/smoke, accessed 31 August 2022), and the CMAQ (Byun and Schere, 2006; Appel et al., 2021). The simulation period covers the

months of June to September 2019, with a 10-day spin-up period in May. Model configurations were similar to our 2016 air quality modeling study (Cheng et al., 2022), with some necessary adjustments for tropospheric dynamics options based on our sensitivity tests for the 2019 study period.

WRF version 4.3.1 (https://github.com/wrf-model/WRF/releases/v4.3.1, accessed 31 August 2022) was used to provide meteorological inputs on a 12-km domain with 471×311 grid cells covering the CONUS (Fig. 1). The atmosphere was

divided into 56 vertical layers with varying thicknesses extending from the surface to 50 hPa, wherein 18 model layers are arranged below 1.5 km, and the lowest (surface) layer has an approximately 10 m midpoint (Table 1). The entire simulation period was broken into overlapping 5.5-day run segments: the first 12 hours of each run segment were discarded because



they were primarily for initializing model fields; the remaining 5 days were used as input for emission processing and air quality simulations. WRF initial and lateral boundary conditions were prepared using the North American Mesoscale

Forecast System (NAM) analysis and 3-hourly forecast (https://www.ncei.noaa.gov/products/weather-climate-models/north-american-mesoscale, accessed 31 August 2022). The main physics, analysis nudging, and dynamics options used in the WRF simulation are summarized in Table 2. Note that the analysis nudging was only performed above the PBL height (or ~1.5 km, whichever is higher) to preserve the nocturnal low-level jet (LLJ), a crucial PBL phenomenon for long-range transport of air pollutants at night (Odman et al., 2019). Also, upper level and vertical velocity damping were turned off to minimize the

impact of numerical filters on stratospheric ozone intrusion, which was estimated to account for approximately 10% of the tropospheric ozone budget (Fusco and Logan, 2003; Liang et al., 2009; Kuang et al., 2012).

SMOKE version 4.7 (https://github.com/CEMPD/SMOKE/releases/SMOKEv47_Oct2019, accessed 06 September 2022) was used to prepare gridded, speciated, hourly anthropogenic emissions for subsequent CMAQ simulations. Because the collaborative 2019 emission modeling platform (EMP) (https://www.epa.gov/air-emissions-modeling/2019-emissions-

modeling-platform, accessed 06 September 2022) was under development at the beginning of this study, the 2016v1 EMP (https://www.epa.gov/air-emissions-modeling/2016v1-platform, accessed 06 September 2022) was used as the base-year inventory and projected to 2019. Note that no growth factor was set for this future-year emission processing. More accurate anthropogenic emissions are expected after the release of the 2019 EMP. Point source emissions were processed in in-line modes. Biogenic emissions were generated in-line in CMAQ using BEIS version 3.6.1 (Bash et al., 2016).

CMAQ version 5.3.3 (https://github.com/USEPA/CMAQ/releases/CMAQv5.3.3_17Aug2021, accessed 06 September 2022) was used to perform two air quality simulations on the EPA 12US2 grid, a 12-km horizontal grid spacing with 396x246 grid cells covering the CONUS (Fig. 1). One is the control simulation (labeled as CNTRL) which was configured with the third revision of the Carbon Bond version 6 (CB6r3) chemical mechanism (Luecken et al., 2019) and the AERO7 aerosol module (Appel et al., 2021). Other science options are listed in Table 3. Note that none of the three CMAQ in-line $LNO_x$ emission

schemes (mNLDN, hNLDN, and pNLDN) was applied in the CNTRL simulation. The other is the lightning simulation (labeled as LGTNO) which added the GLM-based three-dimensional $LNO_x$ emission on top of the CNTRL. Chemical initial and boundary condition input files were extracted and speciated from the Community Atmosphere Model with Chemistry (CAM-chem; Buchholz et al., 2019; Emmons et al., 2020) outputs (https://www.acom.ucar.edu/cam-chem/cam-chem.shtml, accessed 06 September 2022).

## 4 Results and discussions

### 4.1 Contribution of $LNO_x$ to total $NO_x$ emissions

The amount of $NO_x$ emission from lightning, anthropogenic, and soil sources over the model domain was first quantified. As shown in Table 4, lightning flashes produced about $12.43 \times 10^9$ moles $NO_x$ (or equivalently 0.174 Tg N; 1 Tg = $10^{12}$ g) from June through September 2019. The percentage contribution of $LNO_x$ to total $NO_x$ emissions is 12–13% in the summer



months (i.e., June, July, and August), 8% in September, and an average of 11.4% during the study period. These numbers are within the uncertainty range given in previous studies (Bond et al., 2001; Zhang et al., 2003; Schumann and Huntrieser, 2007; Murray, 2016; Kang and Pickering, 2018; Kang et al., 2019a) but are closer to the lower end of the range. For instance, using five years (1995–1999) of NLDN data, Bond et al. (2001) estimated that lightning activity produced approximately 0.323 Tg N over the CONUS in the four-month period from June to September, which is nearly two times the number estimated in this

study (0.174 Tg N). This difference can be attributed to the $LNO_x$ production rate assumption: Bond et al. (2001) used an average production rate of ~400 moles per flash ($6.7 \times 10^{26}$ and $6.7 \times 10^{25}$ $NO_x$ molecules per CG and IC flash, respectively; 29% of flashes are CG), but an *average* production rate $\bar{P}$ of 250 moles $NO_x$ per flash was assumed in this study. Despite the difference in the amount of $LNO_x$ emission, the contribution of the lightning source to the $NO_x$ budget obtained in this study is consistent with what was indicated by Bond et al. (2001). Their results showed that lightning accounts for 11–14% of total

$NO_x$ emissions in the summer months and 5% in September, similar to the percentages summarized in Table 4. However, the estimates by Kang and Pickering (2018) indicated a higher $LNO_x$ contribution to total $NO_x$ emissions, with about 20% for the summer months of 2011 and 10% for September 2011.

The spatial distribution of monthly flash density derived from GLM data is presented in Fig. 2. In the summer months, consistently high flash density was observed in the southeast U.S., especially in Florida, along the Gulf Coast, and the East

Coast. A significant number of lightning strikes also occurred in other regions, including the southern, central, and midwestern U.S. and northwestern Mexico (to the south of Arizona and New Mexico), where the temporal variability of lightning activity was much higher. In September, the frequency of lightning decreased dramatically in the southeast U.S., while Iowa and adjacent states experienced a large number of lightning events. Similar spatial patterns of flash density were presented in a previous long-term lightning climatology study by Holle et al. (2016). Note that they reported lower flash

density values than in this study. This is because Holle et al. (2016) only used CG flashes to compute monthly flash density, while GLM observed both CG and IC flashes.

Figure 2 also presents the spatial distribution of monthly total $NO_x$ emissions from lightning, anthropogenic, and soil sources. Similar to flash density, the amount of $NO_x$ emitted from the lightning source varies significantly with time and location. $LNO_x$ emission is generally greater in the southeastern, southern, central, and midwestern U.S. and northwestern Mexico.

Monthly $LNO_x$ emission in these regions can reach $0.5 \times 10^6$ moles per model grid cell (12 km × 12 km) or higher. However, this is lower than those reported by several recent studies, including Kang and Pickering (2018) and Kang et al. (2019a, 2020), which used a greater $LNO_x$ production rate (350 moles per flash) compared to the mean value used in this study (i.e., $\bar{P} = 250$ moles per flash). On the other hand, the magnitude and the spatial distribution of anthropogenic and soil $NO_x$ emissions are consistent with our 2016 air quality modeling study (Cheng et al. 2022) and Kang and Pickering (2018). In

addition, the contribution of lightning to total $NO_x$ emissions is more significant in the western U.S. and over the water, where anthropogenic $NO_x$ emission is limited.



## 4.2 Impact of LNO$_x$ emission on ground-level ozone and NO$_x$ concentrations

To demonstrate the impact of LNO$_x$ emission on ground-level air quality, mean differences in ground-level ozone, NO$_x$, and
NO$_y$ mixing ratios between two model runs were compared for the entire simulation period (Fig. 3). Ground-level ozone
increase was about 0.5 ppb (1.5%) in the southeast U.S., where lightning activity is intense (Fig. 2a). However, the most
significant ground-level ozone enhancement (~1.0 ppb or 3%) was captured in New Mexico, Arizona, and northwestern
Mexico. This is likely because LNO$_x$ emission accounted for up to 75% of total NO$_x$ emission in this area, much higher than
in the southeast U.S. (Fig. 2e). Unlike ozone, ground-level NO$_x$ concentration slightly decreased in the eastern U.S. The
reason is that NO$_x$ is not chemically conserved (NO$_x$ is converted into NO$_z$ species when producing ozone). In contrast, the
summation of all reactive nitrogen species, NO$_y$, is conserved if only gas-phase reactions are considered and surface loss is
ignored. Therefore, adding LNO$_x$ emission into the LGTNO simulation increased ground-level NO$_y$ mixing ratios, which
showed a similar spatial pattern as ozone.

Model-predicted ground-level ozone and NO$_x$ concentrations were also compared to observations from the EPA Air Quality
System (AQS; https://www.epa.gov/aqs, accessed 24 November 2022). The commonly-used evaluation metrics, including
mean bias (MB), normalized mean bias (NMB), root mean square error (RMSE), normalized mean error (NME), and
correlation coefficient (R), were computed using the Atmospheric Model Evaluation Tool (AMET; Appel et al., 2011;
https://www.epa.gov/cmaq/atmospheric-model-evaluation-tool, accessed 10 August 2022). Because lightning exhibits a
substantial spatial and temporal variation (Kang and Pickering, 2018), the analysis was compiled for the entire model
domain and different geographic regions shown in Fig. 4. The analysis regions follow Kang et al. (2019b) so that regional
statistics obtained in this study can be compared to their results.

Tables 5 and 6 present statistics of maximum daily 8-hour average (MDA8) ozone and daily mean NO$_x$ for August 2019,
respectively, when the percentage contribution of LNO$_x$ emission to total LNO$_x$ emissions was the greatest among the
simulation periods (Table 4). One caveat is that the statistical behavior discussed below may differ for other months because
the prediction skill varies by month. Details on model performance for June, July, and September 2019 are provided in the
supplementary material (see Tables S1–S6). Generally speaking, the impact of LNO$_x$ emission on ground-level ozone and
NO$_x$ was insignificant when averaged on a monthly scale. The difference in monthly mean concentrations was below 1 ppb
(or 2% of the mean observed value) for MDA8 ozone and nearly negligible for daily mean NO$_x$. This is because most of the
NO$_x$ emission from lightning activity happens in the middle and upper troposphere. Only a small portion of LNO$_x$ emission
is released near the surface. Some recent studies (e.g., Kang et al., 2019b, 2020) also indicated that the average impact of
LNO$_x$ emission on ozone and NO$_x$ is small at the ground level.

As shown in Table 5, the CNTRL simulation had slightly better MDA8 ozone statistics than the LGTNO for August 2019 in
the northeast (NE), southeast (SE), Upper Midwest (UM), and Lower Midwest (LM), where the model over-predicted
ground-level ozone concentrations. The situation was reversed in the Rocky Mountains (RM) and Pacific Coast (PC):
ground-level ozone was underestimated in these regions, and statistics of the LGTNO simulation were slightly improved.



This behavior indicates that the extra $NO_x$ produced by lightning promotes ozone formation (unless the environment is VOC-limited, which is often the case in urban areas), increasing ozone biases when over-predicted and reducing when under-predicted. In addition, because lightning activity was prevalent in the SE and RM, changes in the mean bias and error were most significant in these two regions (Fig. 2). Table 6 demonstrates that ground-level $NO_x$ mixing ratios were underestimated in most regions. Changes to the mean ground-level $NO_x$ bias and error due to $LNO_x$ emission at AQS sites were on the order of 0.1 ppb (or 0.1% after normalization), and the correlation was nearly unaffected. Despite this, $NO_x$ statistics were marginally degraded in the NE, SE, and LM and improved in the RM, consistent with the performance of ground-level MDA8 ozone.

Figure 5 presents the impact of $LNO_x$ emission on ground-level MDA8 ozone at each AQS site during August 2019. In the CNTRL simulation, ground-level ozone tended to be over-predicted in the eastern U.S. and under-predicted in the western U.S. Adding $LNO_x$ emission to the simulation noticeably affected ozone statistics in the SE and RM. Also, since ground-level ozone was negatively biased in the RM and positively biased in the SE, the LGTNO simulation improved the prediction of ozone concentrations in the RM (especially in Arizona and New Mexico) but degraded in the SE. However, the difference between the absolute MB of the two simulations was below 2 ppb, while the difference could reach up to 4 ppb when the hNLDN scheme was used (Kang et al., 2019b). As mentioned earlier, this study used a lower (average) $LNO_x$ production rate (i.e., $\bar{P} = 250$ moles per flash) than Kang et al. (2019b), which is likely why a lower impact of $LNO_x$ on ground-level air quality was obtained in this study. Since the $LNO_x$ production rate is still highly uncertain, a more accurate estimate of the $LNO_x$ emission will require a proper constraint on the tropospheric $NO_2$ column, which can be addressed in future studies using $NO_2$ observations from the NASA Tropospheric Emissions: Monitoring of Pollution (TEMPO; Zoogman et al., 2017).

### 4.3 Ozone enhancement in the column

Because a large portion of the $LNO_x$ emission takes place in the free troposphere rather than near the surface (Pickering et al., 1998; Ott et al., 2010; Koshak et al., 2014a; Wang et al., 2015; Kang et al., 2019a,b; Wu et al., 2023), which results in ozone production with a longer residence time, it is expected that ozone enhancement due to $LNO_x$ emission is more significant in the middle and upper troposphere than the ground level. To investigate how the $LNO_x$ emission affects ozone concentrations in the column, vertical distributions of monthly mean ozone enhancement were constructed for different regions, including the entire domain, the southeast U.S. (arbitrarily selected 25–40°N, 75–95°W for computation), and Huntsville, AL. The result for August 2019 is presented in Fig. 6 and discussed below, whereas the results for the other months are provided in the supplementary material (see Fig. S4–S6) to indicate the variation for different months. In August 2019, when averaged for the entire domain, $LNO_x$ increased ozone concentration throughout the troposphere, with a maximum percentage enhancement of 2% (or 1.1 ppb) at ~4 km, which was about twice the percentage at the ground level (1%, or 0.3 ppb). The impact of $LNO_x$ emission on tropospheric ozone was more significant in the southeast U.S., where the average ozone enhancement at 4 km was 4.5% (or 2.3 ppb). At Huntsville, AL, a 5.3% (or ~2.6 ppb) ozone increase was simulated at ~3.6



km. However, these numbers are generally lower than in previous studies in which higher $LNO_x$ production rates were implemented (e.g., Wang et al., 2015; Kang et al., 2019b), which is similar to the ground-level performance as discussed in Section 4.2.

Although average ozone enhancement due to $LNO_x$ emission appears to be small, the impact of $LNO_x$ can be much greater in certain instances. This is because the frequency and intensity of lightning vary significantly with time and location. Shortly after a significant lightning event, ozone concentration in the downwind direction could rise substantially. The Huntsville, AL, area was investigated to demonstrate the details of such scenarios.

The Rocket-city $O_3$ Quality Evaluation in the Troposphere (RO₃QET) lidar (Kuang et al., 2011, 2013), one of the eight systems of the Tropospheric Ozone Lidar Network (TOLNet; https://tolnet.larc.nasa.gov/, accessed 17 January 2023), is located on the campus of the University of Alabama in Huntsville. The RO₃QET is an ozone differential absorption lidar (DIAL) that operates at 289 and 299 nm wavelengths. It can provide continuous observations of ozone profiles below ~10 km at a typical temporal resolution of 10 min with an uncertainty less than 10% (Kuang et al., 2011, 2013).

By examining all available lidar measurements during the 2019 study period, it was realized that better temporal coverage was available in August. A model-to-lidar comparison was performed for all lidar operational periods in August 2019, and the results are presented in Fig. 7. One caveat is that optically thick aerosol layers were present on some days. Previous studies (e.g., Kuang et al., 2011, 2013, 2017) pointed out that heavy aerosol loading can strongly reduce lidar signal-to-noise ratios, resulting in degraded ozone retrievals. Therefore, caution should be taken when interpreting the results of the model-to-lidar comparison under such situations. Since lidar has a high vertical and temporal resolution, it can capture ozone gradients that the model may miss. Despite this, the pattern of model-simulated ozone concentrations was consistent with lidar measurements on most days, suggesting model outputs can adequately represent the state of the atmosphere. During the investigated period, $LNO_x$ emission caused significant (~10 ppb or more) ozone enhancements in the middle and upper troposphere on 12, 13, 19, 21, and 22 August 2019.

After taking a closer look at the difference between model-simulated and lidar-observed ozone mixing ratios, the 19–23 August 2019 period was chosen for further investigation. Figure 8 presents resolution-matched ozone profiles during this period. Lidar measurements were processed vertically to obtain averaged values for each model layer. Also, for each hour during which the lidar made multiple measurements, all 10-min lidar-measured ozone profiles within the hour were averaged. One may notice that model results did not always agree with the lidar observations. This is likely because model simulations were off by one hour or so in time (or one grid cell or two in space). For example, at 1300 UTC on 20 August 2019, the model did a fair job in the lower atmosphere and around 6 km but overpredicted ozone near 4–5.5 km and above ~7 km. In the next few hours, lidar observations indicated a 10–25 ppb ozone increase in the middle and upper troposphere, but the model did not show a significant temporal variation. As a result, model-simulated ozone agreed with lidar at 1500 and 1600 UTC, suggesting model predictions represented an air mass approximately two hours ahead of the observation.

Among the hours presented in Fig. 8, the most significant tropospheric ozone enhancement due to $LNO_x$ occurred at 1600 UTC on 21 August 2019, with an increase of 11.8 ppb at ~4.7 km. To trace the source of this enhancement, NOAA's Hybrid



Single-Particle Lagrangian Integrated Trajectory (HYSPLIT) model (Stein et al., 2015; https://www.ready.noaa.gov/HYSPLIT.php, accessed 19 January 2023) was executed to perform backward trajectory analysis. As shown in Fig. 9a and 9c, some lightning activity was observed near the boundary of Illinois and Kentucky at ~2000 UTC on 20 August 2019. The emitted $LNO_x$ is mixed with the surrounding air when traveling southeastward. This results in increased ozone production in the airmass during daylight hours. As the ozone-enhanced plume reached the Huntsville area after 20-hour transport, ozone concentration increased by more than 10 ppb in the middle troposphere.

During the 2019 study period, the only field campaign providing ozone measurements was the Fire Influence on Regional to Global Environments and Air Quality (FIREX-AQ; https://www-air.larc.nasa.gov/missions/firex-aq/, accessed 23 January 2023). NASA Langley airborne High Spectral Resolution Lidar (HSRL; https://airbornescience.nasa.gov/instrument/HSRL, accessed 29 January 2023), carried by the NASA DC-8 instrument payload, actively remote sensing ozone and other species in the zenith and nadir directions along the flight path. A preliminary analysis indicated that, during the deployment days, lightning activity with more than 10 ppb ozone enhancement was identified on 21, 23, and 26 August 2019 (see Fig. S7–S9 in the supplementary material). In particular, Fig. S7 shows up to 15 ppb ozone enhancements due to $LNO_x$ on August 21.

Since the significant lightning events are limited to a relatively small area within a short time period, the ozone enhancement caused by $LNO_x$ emission can also be limited in time and space. This means that such enhancements can be significant, but may not be evident when averaged over much larger region and longer time. Thus, here we examine the maximum model-simulated tropospheric ozone enhancement caused by LNOx emission. As demonstrated by Fig. 10, within the whole model domain, several regions showed ~40 ppb difference in ozone mixing ratio during the study period, most of which were over water bodies. The maximum ozone enhancement over the continental U.S. was ~38.6 ppb, which occurred at 2100 UTC on 29 June 2019 at 29.970°N, 94.586°W (located between Houston, TX, and Beaumont, TX). Performing backward trajectory analysis suggested that this significant ozone difference had two sources: (1) long-distance chemical transport and (2) lightning activity close to the event. Interestingly, this case was associated with the outflow boundary ahead of a southwestward-moving mesoscale storm (https://www.wpc.ncep.noaa.gov/dailywxmap/index_20190629.html, accessed 25 January 2023).

As illustrated by Fig. 11d, prior to 0300 UTC on 29 August 2019 (2200 CDT on 28 August 2019, local time) background ozone and $NO_x$ in the upwind direction were higher in the LGTNO model run than the CNTRL. This is perhaps due to the prior $LNO_x$ emissions in the LGTNO simulation that causes approximately 5–10 ppb of the ozone difference. The air mass altitude increases as it moves toward Houston, TX, and fresh $LNO_x$ after this time (Fig. 11c) leads to another ~30 ppb ozone increase (Fig. 11d) by the time it is above Houston. Fig. 10c indicates $LNO_x$ emission over southwestern Arkansas and northwestern Louisiana after midnight and in southeastern Texas in the morning. The time series in Fig. 11 indicate that NO was first produced by lightning at night. Then, since there was no sunlight, the emitted NO was almost instantly oxidized by ozone and converted to $NO_2$. This is evident from the sharp $NO_2$ increase in Fig. 11g and the corresponding ozone reduction in Fig. 11d. Shortly after sunrise, due to photochemistry, ozone concentration starts to increase. Photochemical activity and the injection of additional $LNO_x$ along the trajectory leads to significant ozone increase (38.6 ppb more than the CNTRL). In





addition, surface insolation drops dramatically at the time of LNO$_x$ emission during the day, suggesting that the model correctly produced clouds at locations where lightning flashes were observed.

An interesting feature in this trajectory is the chemical evolution of the air mass with respect to its location and the role of

atmospheric dynamics (Parrish et al., 2012). Figure 11h shows a rapid increase in formaldehyde after sunrise up to 1500 UTC. This increase is positively correlated with NO and negatively correlated with NO$_2$, indicating the presence of adequate VOC and a very active photochemistry. The elevation of air mass is more than 5 km during this period. Thus, the VOC must have been transported from near surface pollution in the Houston area. After 1500 UTC, HCHO starts to decrease, while ozone continues to increase. The timing of the decrease coincides with the injection of fresh lightning NO. This is typical

behavior of a NO$_x$-limited air mass. From the time-series in Fig. 11, it can be deduced that prior to 1500 UTC as the clouds are forming, vertical transport of boundary-layer air to higher altitudes, increases VOC and creates a NO$_x$-limited chemical environment. This is evident by the decrease in NO$_x$, increase in HCHO, increase in relative humidity, and relatively lower surface insolation. However, after 1500 UTC, with the injection of fresh LNO$_x$ in this NO$_x$-limited air mass, rapid ozone production transpires. The rapid ozone production is being helped by the fact that at this time the air mass is higher up in the

clouds and perhaps exposed to relatively higher actinic flux (Ryu et al., 2017).

## 5 Conclusions

This study is our first attempt to employ the LNO$_x$ emission estimates derived from GLM space-borne lightning observations in air quality model simulations. Our results showed that, for the CONUS domain, lightning activity released approximately 0.174 Tg N of NO$_x$ into the atmosphere between June and September 2019, accounting for 11.4% of the total NO$_x$ budget

over this area. Performing two CMAQ simulations revealed that adding the GLM-based LNO$_x$ emission could increase ozone concentration by a domain-wide average of 1–2% (or 0.3–1.5 ppb), with the maximum enhancement at ~4 km above ground level and the minimum near the surface. The strength and frequency of lightning events are not evenly distributed across the CONUS, and so is the impact of LNO$_x$ emission on ozone concentration. Due to relatively more lightning and biogenic VOC in the southeast U.S., this region exhibited the most significant difference in ground-level ozone, with up to 1

ppb (or 2% of the mean observed value) increase for MDA8 ozone. However, although the numbers above generally fall within the uncertainty range given in previous studies, many are closer to the lower bound. This is due to using a smaller average LNO$_x$ production rate (i.e., $\bar{P} = 250$ moles NO$_x$ per flash) in estimation of β in this study compared to other recent studies (Kang et al., 2019a, 2020). It is important to note that although this work assumes a fixed value of the *average* LNO$_x$ production rate per flash, the β method employed still assigns *distinct* LNO$_x$ production values to each flash in general, based

directly on the variable/unique GLM flash optical energy observations.

While the average influence of LNO$_x$ on tropospheric ozone over the entire study period was small, the local impact on shorter time scale could be considerable. The LGTNO simulation at Huntsville, AL agreed with the hourly averaged ozone lidar observations in general, despite some discrepancies due to the different temporal resolutions. The results of backward



trajectory analyses illustrated that long-range chemical transport and upwind lightning activity are the two major
contributing factors for significant ozone enhancement. A case study was presented, exhibiting a tropospheric ozone enhancement of 38.6 ppb over Houston, TX. Trajectory analysis demonstrated that during the formation of storms, boundary layer air that is rich in VOC can be transported to higher altitudes and diluted to create a $NO_x$-limited environment. In such an environment, addition of fresh NO from lightning can lead to significant ozone production. Furthermore, storms provide a mechanism for the transport of higher tropospheric $LNO_x$ to the surface and the transport of boundary layer air to higher
altitudes.

Potential improvements are expected in future studies after making proper adjustments. As indicated, the average $LNO_x$ emission rate in this study is on the lower end of the estimates and needs to be increased for the follow-up studies. Moreover, a more reasonable average $LNO_x$ production rate can be obtained by constraining tropospheric $NO_x$ columns based on geostationary (e.g., TEMPO) satellite observations. Also, implementing cloud assimilation techniques can reduce the
temporal and spatial discrepancy between model-simulated clouds and GLM-captured lightning flashes.

**Acknowledgments**. The present research was originally conducted as part of the author's Ph.D. dissertation work at the University of Alabama in Huntsville. The findings presented here were accomplished under partial support from NASA Science Mission Directorate Applied Sciences Program (NASA Grant 80NSSC18K1598). In addition, a portion of the work
by co-author Koshak was supported by the Precipitation and Lightning Work Package for the Internal Science Funding Model (ISFM) project Lightning as an Indicator of Climate under NASA Headquarters (Dr. Jack Kaye and Dr. Lucia Tsaoussi), that in part supports NASA's participation in the National Climate Assessment (NCA). Koshak's work on this effort pertaining to GLM-16/17 data was supported by the NOAA GOES-R Series Program (Calibration and Algorithm Working Groups) under Drs. Dan Lindsey and Jaime Daniels. Shi Kuang's work is supported by NASA's TOLNet program.
Note that the results in this study do not necessarily reflect policy or science positions by the funding agencies.

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




**Table 1: Model vertical layers and their approximate height above ground level.**

| Layer | Sigma | Pressure [hPa] | Height [m] | Thickness [m] |
|---|---|---|---|---|
| Surface | 1.0000 | 1000.0 | 0.0 | — |
| 1 | 0.9975 | 997.6 | 20.9 | 20.9 |
| 2 | 0.9950 | 995.3 | 41.8 | 20.9 |
| 3 | 0.9920 | 992.4 | 66.9 | 25.1 |
| 4 | 0.9880 | 988.6 | 100.5 | 33.6 |
| 5 | 0.9830 | 983.9 | 142.7 | 42.2 |
| 6 | 0.9780 | 979.1 | 185.1 | 42.4 |
| 7 | 0.9730 | 974.4 | 227.6 | 42.5 |
| 8 | 0.9660 | 967.7 | 287.4 | 59.8 |
| 9 | 0.9580 | 960.1 | 356.2 | 68.8 |
| 10 | 0.9490 | 951.6 | 434.1 | 77.9 |
| 11 | 0.9390 | 942.1 | 521.4 | 87.3 |
| 12 | 0.9270 | 930.7 | 627.1 | 105.7 |
| 13 | 0.9140 | 918.3 | 742.8 | 115.7 |
| 14 | 0.9000 | 905.0 | 868.9 | 126.1 |
| 15 | 0.8850 | 890.8 | 1005.7 | 136.8 |
| 16 | 0.8690 | 875.6 | 1153.7 | 148.0 |
| 17 | 0.8530 | 860.4 | 1303.8 | 150.1 |
| 18 | 0.8370 | 845.2 | 1456.1 | 152.3 |
| 19 | 0.8210 | 830.0 | 1610.7 | 154.6 |
| 20 | 0.8050 | 814.8 | 1767.7 | 157.0 |
| 21 | 0.7870 | 797.7 | 1947.2 | 179.5 |
| 22 | 0.7680 | 779.6 | 2140.1 | 193.0 |
| 23 | 0.7480 | 760.6 | 2347.3 | 207.1 |
| 24 | 0.7260 | 739.7 | 2580.2 | 232.9 |
| 25 | 0.7020 | 716.9 | 2840.5 | 260.3 |
| 26 | 0.6760 | 692.2 | 3130.2 | 289.8 |
| 27 | 0.6480 | 665.6 | 3452.0 | 321.8 |
| 28 | 0.6200 | 639.0 | 3784.5 | 332.5 |
| 29 | 0.5920 | 612.4 | 4128.6 | 344.0 |
| 30 | 0.5640 | 585.8 | 4485.1 | 356.5 |
| 31 | 0.5360 | 559.2 | 4855.0 | 370.0 |



| 32 | 0.5080 | 532.6 | 5239.7 | 384.6 |
|---|---|---|---|---|
| 33 | 0.4810 | 507.0 | 5625.7 | 386.0 |
| 34 | 0.4550 | 482.3 | 6012.7 | 387.0 |
| 35 | 0.4290 | 457.6 | 6416.1 | 403.4 |
| 36 | 0.4040 | 433.8 | 6820.9 | 404.8 |
| 37 | 0.3790 | 410.1 | 7244.0 | 423.1 |
| 38 | 0.3550 | 387.3 | 7669.2 | 425.2 |
| 39 | 0.3330 | 366.4 | 8076.9 | 407.8 |
| 40 | 0.3120 | 346.4 | 8483.9 | 407.0 |
| 41 | 0.2920 | 327.4 | 8889.3 | 405.4 |
| 42 | 0.2730 | 309.4 | 9292.1 | 402.8 |
| 43 | 0.2540 | 291.3 | 9714.0 | 421.9 |
| 44 | 0.2350 | 273.3 | 10157.0 | 443.1 |
| 45 | 0.2160 | 255.2 | 10623.8 | 466.8 |
| 46 | 0.1970 | 237.2 | 11117.2 | 493.4 |
| 47 | 0.1780 | 219.1 | 11640.9 | 523.7 |
| 48 | 0.1590 | 201.1 | 12199.2 | 558.3 |
| 49 | 0.1400 | 183.0 | 12797.6 | 598.5 |
| 50 | 0.1200 | 164.0 | 13478.7 | 681.1 |
| 51 | 0.1000 | 145.0 | 14222.8 | 744.1 |
| 52 | 0.0800 | 126.0 | 15044.5 | 821.7 |
| 53 | 0.0600 | 107.0 | 15964.4 | 919.9 |
| 54 | 0.0400 | 88.0 | 17013.3 | 1048.9 |
| 55 | 0.0200 | 69.0 | 18240.2 | 1226.9 |
| 56 | 0.0000 | 50.0 | 19731.7 | 1491.5 |



**Table 2: WRF physics, analysis nudging, and dynamics options.**

|  | Option | Setting |
|---|---|---|
| Physics | Microphysics | Morrison 2-moment scheme (Morrison et al., 2009) |
|  | Cumulus | Multiscale Kain-Fritsch (Zheng et al., 2016) |
|  | Radiation | RRTMG (Iacono et al., 2008) |
|  | Surface layer | Pleim (Pleim, 2006) |
|  | Land surface model (LSM) | Pleim-Xiu (Xiu and Pleim, 2001; Pleim and Xiu, 2003) |
|  | Planetary boundary layer (PBL) | ACM2 (Pleim, 2007a,b) |
|  |  |  |
| Analysis nudging | Nudging height cutoff | Above the PBL or the ~1.5-km model layer, whichever is higher |
|  | $U$, $V$ nudging coefficient | $3.0 \times 10^{-4}$ s$^{-1}$ |
|  | $T$ nudging coefficient | $3.0 \times 10^{-4}$ s$^{-1}$ |
|  | $Q$ nudging coefficient | $1.0 \times 10^{-5}$ s$^{-1}$ |
|  |  |  |
| Dynamics | Model dynamics | Non-hydrostatic |
|  | Time integration | Runge-Kutta, third order |
|  | Vertical coordinate | Terrain following |
|  | Turbulence and mixing | Without vertical correction |
|  | Eddy coefficient | Horizontal Smagorinsky, first order |
|  | Sixth order diffusion | Off |
|  | Upper level damping | Off |
|  | Vertical velocity damping | Off |
|  | Advection options | Positive definite |




**Table 3: CMAQ science options**.

| Science Option | Setting |
| --- | --- |
| Gas phase chemistry solver | CB6r3 (Luecken et al., 2019) |
| Aerosol chemistry module | AERO7 (Appel et al., 2021) |
| Dry deposition scheme | M3Dry |
| In-line biogenic emission module | BEIS3 |
| CTM_OCEAN_CHEM | Y |
| CTM_WB_DUST | Y |
| CTM_WBDUST_BELD | BELD3 |
| CTM_LTNG_NO | N |
| KZMIN | Y |
| CTM_MOSAIC | N |
| CTM_FST | N |
| PX_VERSION | Y |
| CLM_VERSION | N |
| NOAH_VERSION | N |
| CTM_ABFLUX | N |
| CTM_BIDI_FERT_NH3 | Y |
| CTM_HGBIDI | N |
| CTM_SFC_HONO | Y |
| CTM_GRAV_SETL | Y |
| CTM_BIOGEMIS | Y |



**Table 4: Total monthly LNO$_x$, anthropogenic NO$_x$, and soil NO emissions of the model domain.**

|  | LNO$_x$ [$\times10^9$ moles] | Anthropogenic NO$_x$ [$\times10^9$ moles] | Soil NO [$\times10^9$ moles] | Total NO$_x$ [$\times10^9$ moles] |
|---|---|---|---|---|
| June | 3.37 (12.3%) | 19.75 (72.3%) | 4.20 (15.4%) | 27.32 |
| July | 3.45 (12.1%) | 20.59 (72.0%) | 4.58 (16.0%) | 28.62 |
| August | 3.65 (12.9%) | 20.57 (72.5%) | 4.15 (14.6%) | 28.37 |
| September | 1.96 (8.0%) | 19.13 (78.1%) | 3.42 (14.0%) | 24.51 |
| Total | 12.43 (11.4%) | 80.04 (73.6%) | 16.35 (15.0%) | 108.82 |



**Table 5: Ground-level MDA8 ozone statistics over the model domain and geographic regions for August 2019. Bold numbers indicate better performance for each case.**

| Region | Case | Record | OBS [ppb] | MOD [ppb] | MB [ppb] | NMB [%] | RMSE [ppb] | NME [%] | R |
|---|---|---|---|---|---|---|---|---|---|
| Domain | CNTRL | 35132 | 44.0 | 45.6 | **1.6** | **3.7** | **8.3** | **14.5** | **0.76** |
|  | LGTNO | 35132 | 44.0 | 46.0 | 2.0 | 4.6 | 8.4 | 14.6 | **0.76** |
| NE | CNTRL | 5518 | 42.1 | 46.2 | **4.2** | **9.9** | **8.2** | **15.2** | **0.77** |
|  | LGTNO | 5518 | 42.1 | 46.5 | 4.4 | 10.6 | 8.3 | 15.4 | **0.77** |
| SE | CNTRL | 5912 | 39.3 | 43.7 | **4.4** | **11.3** | **8.2** | **16.5** | **0.78** |
|  | LGTNO | 5912 | 39.3 | 44.5 | 5.2 | 13.2 | 8.6 | 17.4 | **0.78** |
| UM | CNTRL | 8767 | 42.3 | 43.7 | **1.4** | **3.3** | **6.8** | **12.3** | **0.75** |
|  | LGTNO | 8767 | 42.3 | 43.9 | 1.7 | 4.0 | 6.8 | 12.5 | **0.75** |
| LM | CNTRL | 3477 | 39.7 | 43.6 | **3.9** | **9.8** | **8.9** | **18.1** | **0.79** |
|  | LGTNO | 3477 | 39.7 | 44.1 | 4.4 | 11.0 | 9.1 | 18.3 | **0.79** |
| RM | CNTRL | 5985 | 50.9 | 50.0 | -0.8 | -1.6 | 8.1 | 12.0 | 0.60 |
|  | LGTNO | 5985 | 50.9 | 50.6 | **-0.3** | **-0.5** | **7.8** | **11.5** | **0.63** |
| PC | CNTRL | 5443 | 48.9 | 46.6 | -2.4 | -4.8 | **10.3** | **16.1** | **0.80** |
|  | LGTNO | 5443 | 48.9 | 46.6 | **-2.3** | **-4.8** | **10.3** | **16.1** | **0.80** |






**Table 6: Ground-level daily mean NO$_x$ statistics over the model domain and geographic regions for August 2019. Bold numbers indicate better performance for each case.**

| Region | Case | Record | OBS [ppb] | MOD [ppb] | MB [ppb] | NMB [%] | RMSE [ppb] | NME [%] | R |
|---|---|---|---|---|---|---|---|---|---|
| Domain | CNTRL | 10705 | 8.69 | 7.86 | **-0.83** | **-9.52** | **8.22** | **54.80** | **0.57** |
| | LGTNO | 10705 | 8.69 | 7.86 | -0.83 | -9.53 | **8.22** | **54.80** | **0.57** |
| NE | CNTRL | 1606 | 10.33 | 9.67 | **-0.66** | **-6.38** | **9.56** | **58.90** | **0.50** |
| | LGTNO | 1606 | 10.33 | 9.67 | -0.66 | -6.41 | **9.56** | **58.90** | **0.50** |
| SE | CNTRL | 1002 | 11.90 | 9.42 | **-2.48** | **-20.80** | **9.43** | **51.50** | **0.55** |
| | LGTNO | 1002 | 11.90 | 9.41 | **-2.48** | -20.90 | **9.43** | **51.50** | **0.55** |
| UM | CNTRL | 1167 | 10.40 | 8.26 | **-2.14** | **-20.60** | **7.71** | **44.90** | **0.58** |
| | LGTNO | 1167 | 10.40 | 8.26 | **-2.14** | **-20.60** | **7.71** | **44.90** | **0.58** |
| LM | CNTRL | 1685 | 6.72 | 7.45 | 0.73 | **10.90** | **7.41** | **65.30** | **0.42** |
| | LGTNO | 1685 | 6.72 | 7.44 | **0.73** | **10.90** | **7.41** | **65.30** | **0.42** |
| RM | CNTRL | 2623 | 4.55 | 4.39 | -0.16 | -3.47 | **4.19** | **51.00** | **0.77** |
| | LGTNO | 2623 | 4.55 | 4.39 | **-0.16** | **-3.42** | **4.19** | **51.00** | **0.77** |
| PC | CNTRL | 2663 | 11.00 | 9.63 | **-1.37** | **-12.50** | **10.20** | **55.40** | **0.49** |
| | LGTNO | 2663 | 11.00 | 9.63 | **-1.37** | **-12.50** | **10.20** | **55.40** | **0.49** |




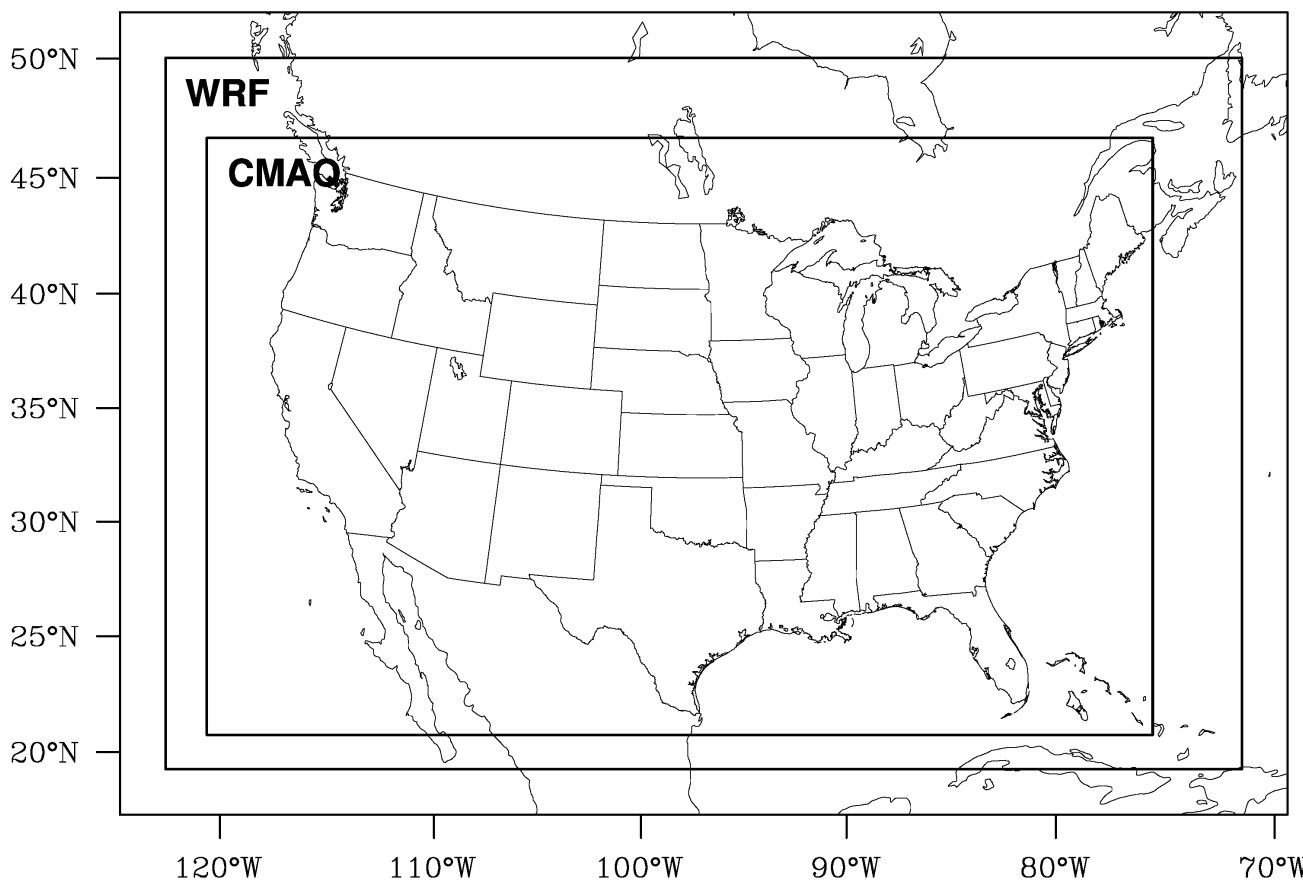

**Figure 1: WRF and CMAQ model domain.**





**Figure 2: Spatial distribution of monthly flash density and NO$_x$ emissions from lightning, anthropogenic, and soil sources for June through September 2019. (a) Total flashes per km$^2$ per month; monthly total NO$_x$ emissions (in 10$^6$ moles) from (b) lightning, (c) anthropogenic, and (d) soil sources per model grid cell (12 km × 12 km, or 144 km$^2$); (e) the ratio of LNO$_x$ to total NO$_x$ emissions.**





**Figure 3: Spatial distribution of mean differences in ground-level ozone, NO$_x$, and NO$_y$ mixing ratios between the LGTNO and the CNTRL simulations for 01 June 2019 through 30 September 2019. (a) Ozone difference, (b) ozone percentage change, (c) NO$_x$ difference, (d) NO$_x$ percentage change, (e) NO$_y$ difference, and (f) NO$_y$ percentage change.**






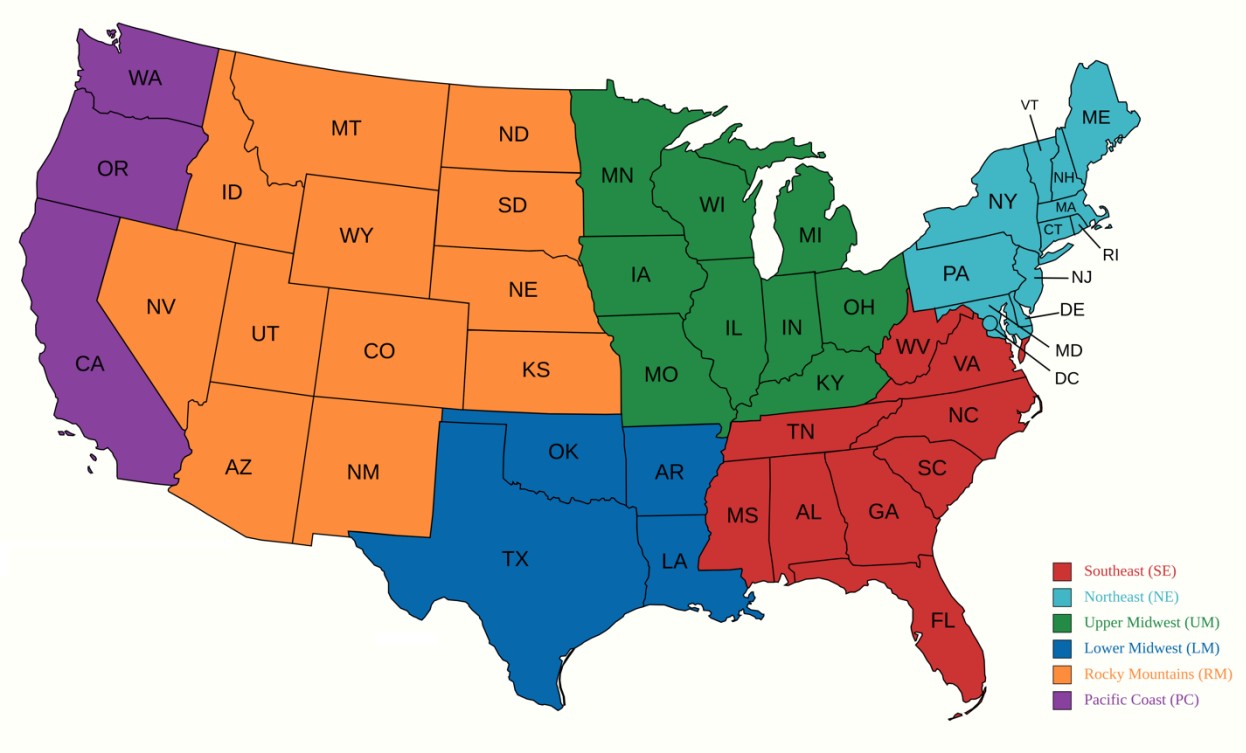

**Figure 4: Geographical regions for statistical analysis.**




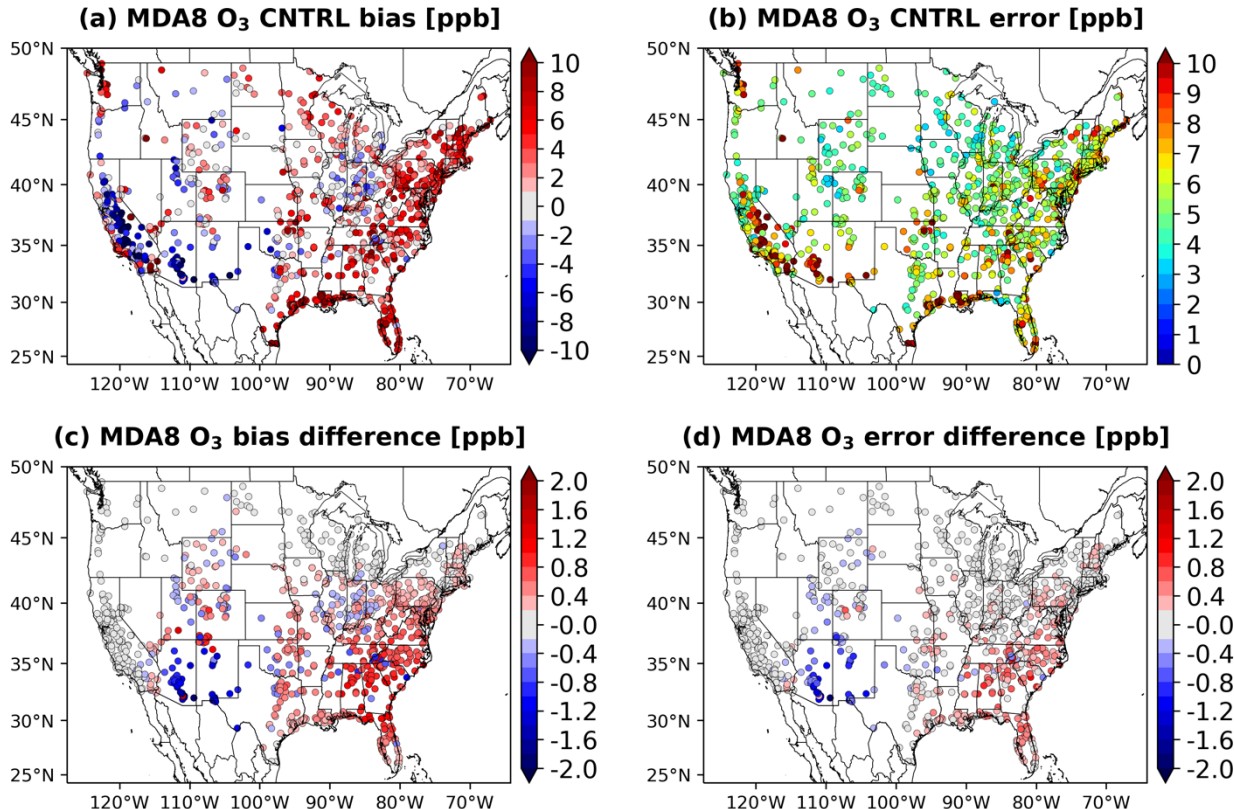

**Figure 5: Spatial distribution of ground-level MDA8 ozone statistics for August 2019. (a) Mean bias of the CNTRL; (b) mean error of the CNTRL; (c) absolute mean bias difference between the LGTNO and the CNTRL; (d) mean error difference between the LGTNO and the CNTRL. In (c) and (d), negative and positive values represent improved and degraded statistics when including the NO$_x$ emission, respectively.**





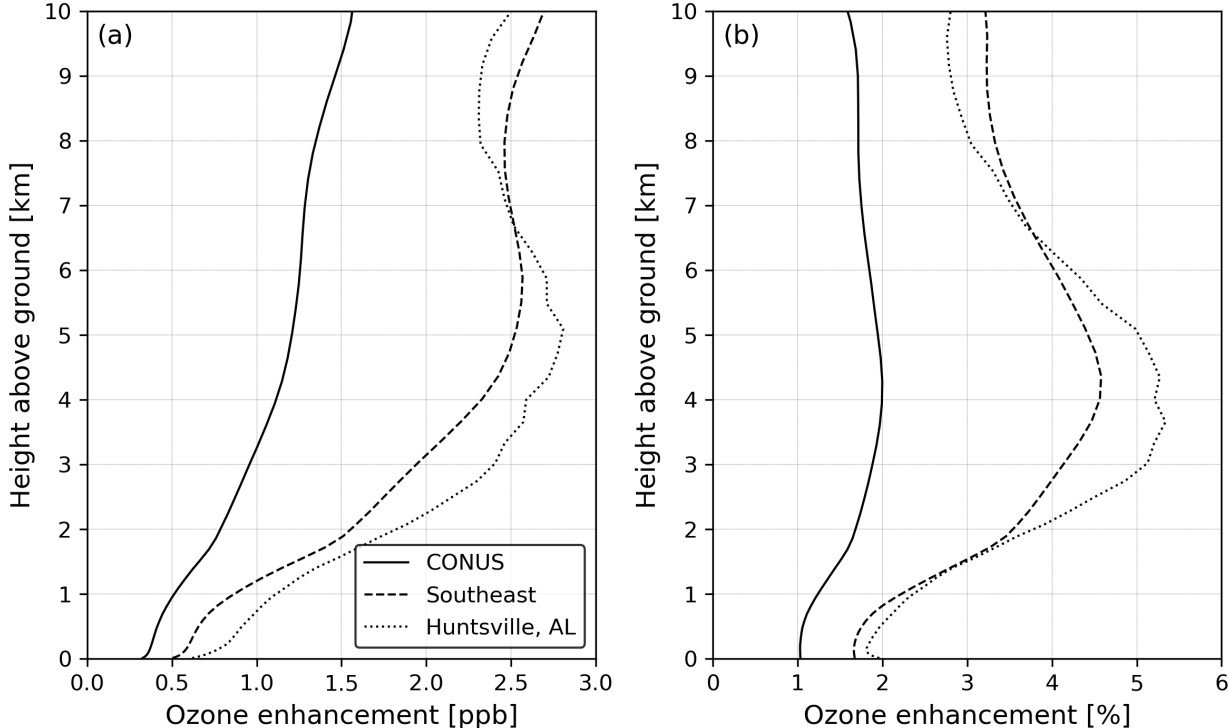

**Figure 6: Vertical distribution of average ozone enhancement due to lightning NO$_x$ emission during August 2019 for the CONUS domain, the southeast U.S. (arbitrarily selected 25–40°N, 75–95°W for computation), and Huntsville, AL. (a) Ozone enhancement in ppb; (b) ozone enhancement in percent.**





**Figure 7: Time-height cross sections of lidar-measured and model-simulated ozone mixing ratio at Huntsville, AL. (a) Lidar-measured ozone profiles; (b) simulated ozone mixing ratio by the CNTRL model run; (c) ozone difference between the LGTNO and the CNTRL; (d) lidar-observed aerosol extinction coefficient at 299 nm. All available lidar data in August 2019 and the corresponding model predictions are presented.**





**Figure 8: Hourly mean lidar-measured and model-simulated ozone profiles at Huntsville, AL, for all lidar observation periods between 19 and 23 August 2019. Black lines represent lidar observations after being averaged hourly and vertically to match the model resolution. Shaded regions indicate ranges of lidar measurements within each hour for each model layer. The red and blue lines represent model predictions of the CNTRL and the LGTNO, respectively.**



**Figure 9: Backward trajectory analysis of the air mass arrived at 34.724°N, 86.645°W (Huntsville, AL) at ~4.7 km above ground level at 1600 UTC on 21 August 2019. (a) Latitude and longitude, (b) parcel height, (c) hourly lightning NOₓ emission, (d) ozone difference (between the LGTNO and the CNTRL), (e) NOₓ difference, (f) NO difference, (g) NO₂ difference, (h) HCHO difference, (i) surface insolation, (j) relative humidity, and (k) air temperature along the trajectory. The squares indicate 6-hour intervals along the trajectory. The star indicates the ending point of the trajectory. Shaded regions in the time series plots indicate local nighttime hours.**



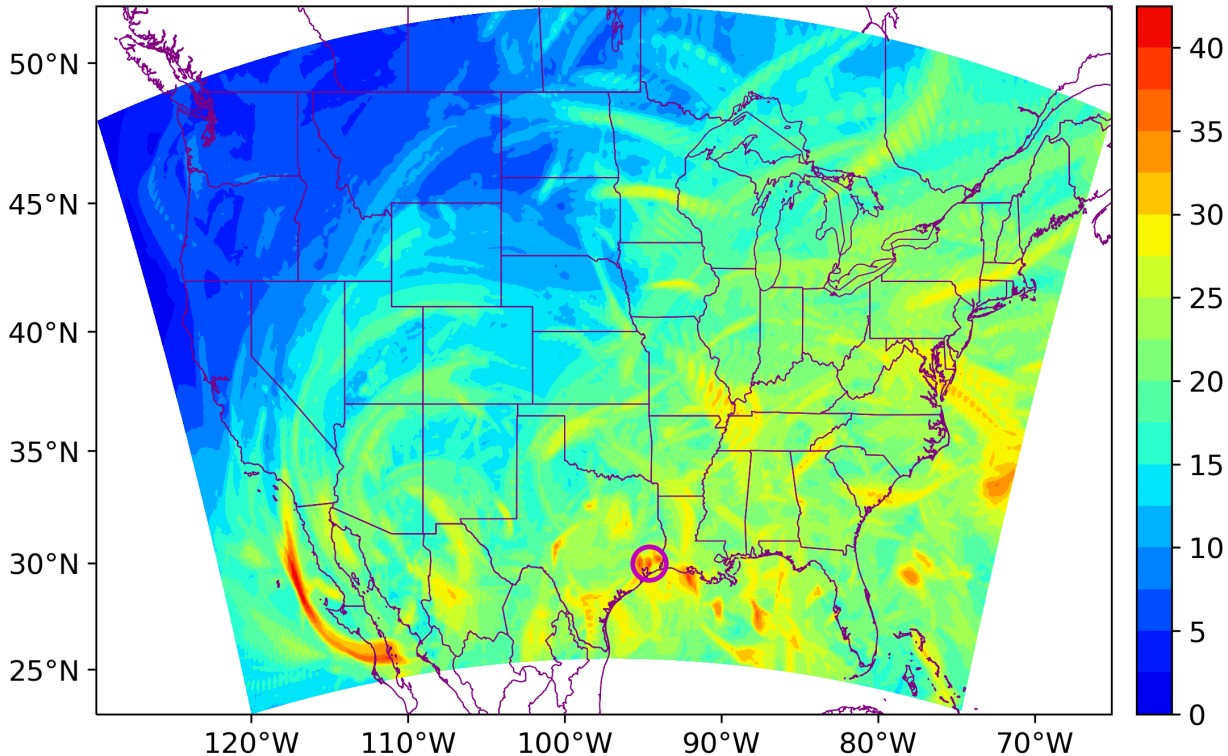

**Figure 10: Spatial distribution of maximum ozone enhancement within the troposphere due to LNO$_x$ emission. The case of interest showing a ~38.6 ppb ozone increase occurred at 2100 UTC on 29 June 2019 at 29.970°N, 94.586°W (located between Houston, TX, and Beaumont, TX, highlighted by the magenta circle) at ~5.9 km above ground level.**

770





**Figure 11:** Backward trajectory analysis of the air mass arrived at 29.970°N, 94.586°W (located between Houston, TX, and Beaumont, TX) at ~5.9 km above ground level at 2100 UTC on 29 June 2019. (a) Latitude and longitude, (b) parcel height, (c) hourly lightning NOₓ emission, (d) ozone difference (between the LGTNO and the CNTRL), (e) NOₓ difference, (f) NO difference, (g) NO₂ difference, (h) HCHO difference, (i) surface insolation, (j) relative humidity, and (k) air temperature along the trajectory. The squares indicate 6-hour intervals along the trajectory. The star indicates the ending point of the trajectory. Shaded regions in the time series plots indicate local nighttime hours.