# Peer review of "Utility of Geostationary Lightning Mapper Derived Lightning NOx Emission Estimates in Air Quality Modeling Studies"

_EGUsphere, 2023_

## Author Response (AR1)

**Responses to Reviewers.**

**Response to Reviewer #1:**

Summary: Well-written easy-to-read manuscript

Major Comments: None

Minor Comments:

While not necessary, it would be interesting to include a plot showing the impact of LNOx on particulate matter concentrations, especially nitrate concentrations.

**Response**: We thank the reviewer for the comments. Indeed, it would have been interesting to include such a plot. However, due to the focus of this study on gases, we didn't save particulate matter concentrations at the time of investigation (due to the storage limits). However, some comments have been added at the end of the conclusion section as this is worth investigating in future studies.

We also agree with the reviewer that we need to be consistent in the paper in using LNO (lightning-generated NO) and LNOx (lightning-generated NOx). In fact, many authors assume that the readers are aware of this distinction and recognize that lightning produces NO, but due to the fast oxidation in the atmosphere, atmospheric measurements should consider the sum of NO and NO2. We thank the reviewer for bringing this to our attention. The paper is revised to use LNO throughout the paper when talking about lightning-generated NO.

L10: For consistency, I would use nitric oxide throughout.

**Response**: Nitric oxide was used consistently.

L16: ppb --> ppbv

**Response**: This was corrected.

L20: Not necessary to speculate that your values are too low

**Response**: We thank the reviewer for the comment. Based on our analysis, we think our LNO emission is lower than expected, and the 250 moles per flash LNO production rate is likely an underestimate. But verifying this speculation requires further investigation. We revised the sentence to include the decreasing trend of anthropogenic emissions and to state the need for further refinements (without speculating on an increase or decrease).

L129: How variable is NOx production per flash when this method is used?  What is the coefficient of variation?

**Response**: The per-flash variation of NO production is directly related (controlled) by observed flash energy. While we assume a global average production rate for estimating $\beta$, the per-flash production is calculated from the individual flash optical energy. Since this study investigated the application of GLM-based LNO emission in air quality modeling studies, we decided not to discuss the variations in the LNO

production rate here. In fact, these questions have been discussed thoroughly in Wu et al. (2023) Section 5, particularly Figure 9 (spatial variation) and Figure 11 (temporal variation).

L168: It would be useful to add a plot in the main or supplemental section showing the assumed vertical distribution for the NO emissions that resulted after the application of LNOM.

**Response**: We thank the reviewer for this suggestion. A figure showing monthly LNO emission profiles is now added to the supplemental material.

L168: Does the vertical distribution of emissions vary with the modeled cloud top pressure or is the highest layer with emissions determined using climatological information and not tied to the modeled cloud.

**Response**: Vertical distribution of LNO emission was computed off-line by the LNO emission model. It is purely based on GLM observations and climatological information. Currently is not tied to the modeled cloud. We mentioned this as an uncertainty source later in Section 2.2.

L182: What were the necessary adjustments for tropospheric dynamics? Can the differences in options between this paper and Cheng et al. (2022) be shown in Table 2 and/or Table 3?

**Response**: Thank you for the comment. We added two tables in the supplement materials (Tables S1 and S2) to indicate what options were used for WRF and CMAQ by Cheng et al. (2022). Readers are referred to the supplement material for details. The main reason for adjustments in the current simulations was the increased vertical resolution in the upper troposphere. Increased resolution was necessary to better resolve stratospheric/tropospheric exchange. However, the increased resolution also caused Courant number violation within strong storms. Thus, we had to adjust some of the damping options in WRF to alleviate the problem.

L194-196: Be careful here. Does the stratosphere contribute 10% to the tropospheric ozone budget or do numerical filters contribute 10% to the budget.

**Response**: Thank you for the comment. This has been revised to avoid ambiguity.

L229: Presumably, anthropogenic emissions of NOx have decreased since the Bond et al. study. Does this account for the similar percentages despite lower LNOx emissions?

**Response**: Thank you for this comment. Yes, due to the decrease in anthropogenic NOx emission over the CONUS for the past two decades, one would expect LNO emission accounts for a larger portion of the total NOx budget. This means that the values from Bond et al. (2001) should be scaled up for the study period. In fact, this is one more indication that our beta value needs refinement. We revised the text to indicate this fact.

L231: Did Kang and Pickering assume a mean PE of 350 mols per flash as specified on L247? (2018)?

**Response**: The 350 moles per flash production rate was not explicitly stated in Kang and Pickering (2018). But, 350 moles/flash is explicitly stated in Kang et al. (2019) and set as the default value for both ICs and CGs in CMAQ (Kang et al. 2019). This value is also used by Dr. Kang's recent research (e.g., Kang et al. 2020). Text was revised to include all citations.

L233: Do you make any assumptions wrt detection efficiency when plotting the GLM flash densities?

**Response**: We did not make any adjustments/corrections w.r.t. GLM detection efficiency. The GLM-based LNO emission model (Wu et al. 2023) uses observed optical energy by GLM as shown in equation 2. As described in Wu et al. (2023), GLM-east and GLM-west observations are combined to alleviate the problem of DE with increased viewing angle. Direct output (LNO emission) from the emissions model was used without any modifications. Therefore, we plotted the GLM flash density using exactly all the flashes from combined GLM data.

L257: What role does the altitude of the surface play in the large enhancements out west?

**Response**: Convective storms are more active over the mountains in the west, leading to more lightning events and LNO emissions. In addition, anthropogenic NOx emissions are relatively lower in these areas. As a result, the contribution of LNO to total NOx is relatively larger, and its impact on ground ozone is significant.

L282-286: and Table 5 Consider evaluating the simulations using the centered-RMSE as opposed to the "total" RMSE as the centered-RMSE does not include the contribution of biases to the RMSE.

**Response**: Thank you for the suggestion. We agree that using "centered" RMSE can exclude the contribution of biases to the RMSE. Tables 5 and 6 and those in the supplementary material have been revised.

Table 1: How did you obtain the approximate heights?

**Response**: They were estimated using Equation 1.3 in PSU NCAR Mesoscale Modeling System Tutorial Class Notes and User's Guide: MM5 Modeling System Version 3, page 1-11. This is the approximation we used to set model sigma levels. We decided to present these approximations instead of calculating an average AGH from WRF output (geopotential and terrain height). The table caption was corrected to indicate approximate geopotential height.

Table 5: Replace RMSE with centered RMSE. Refer reader to Figure 4 for interpretation of regions.

**Response**: Thank you for the suggestion. Tables 5 and 6 and those in the supplementary material have been revised to show centered RMSE rather than total RMSE. Also, we have added a sentence to refer the readers to Fig. 4 for the interpretation of regions.

Table 6: Is this for NOx or NO2? Add some information on what you mean by NOx measurements as NO2 is usually measured albeit often with interferences.

**Response**: It is the NOx (= NO + NO2) comparison. NOx observations used here were extracted from EPA AQS network hourly data as part of the AMET-ready observation data from the CMAS Center Data Warehouse (https://www.epa.gov/cmaq/atmospheric-model-evaluation-tool). The AMET-ready data has multiple components, including O3, CO, SO2, NO, NO2, NOx, NOy, PM2.5, PM10, etc. The hourly NOx data were then processed to obtain daily values for use with the AMET. The original pre-generated hourly NOx observation files can be accessed from the AQS data mart (https://aqs.epa.gov/aqsweb/airdata/download_files.html#Raw).

**Response**: The original "error" was the mean error, i.e., mean (abs(obs - model)). But considering the comment regarding centered RMSE, ME subplots have been replaced by centered RMSE for Figure 5 and those in the supplementary material. All corresponding notations and figure captions have been revised to reflect this change.

Minor Grammatical Comments:

L22: for the significant ozone increase --> for significant ozone increases

**Response:** This was corrected.

L44: columnar --> column

**Response:** This was corrected.

L47: study by --> review by

**Response:** This was corrected.

L72: that compensates for some of the uncertainties and converts --> that converts

**Response**: This was corrected.

L89: would likely introduce some uncertainty --> adds uncertainty

**Response**: This was corrected.

L96: from GLM --> from the GLM

**Response**: This was corrected.

L97: distributed at --> distributed via the

**Response**: This was corrected.

L98: GLM Level 2 --> The GLM Level 2

**Response**: This was corrected.

L105: Detection efficiencies also vary with the type of storm. conditions are DEs smaller than 70%

**Response**: Thank you for this comment. The above 70% DE is an average. The sentence has been revised to be more precise.

L139: is biased by the mean LNOx production rate --> varies linearly with the assumed mean LNOx production rate, i.e., this method constrains the geographic distribution of PE but not its magnitude.

**Response**: The sentence was revised.

L286: which is often the case in urban --> which may be the case in urban

**Response**: Was revised.

L368: when averaged over much larger region and longer time --> when averaged over much larger regions and/or times.

**Response**: Was corrected.

Figure 9: air mass arrived --> air mass that arrived

**Response**: Was corrected.

**Response to Reviewer #2**:

Major Comments:

This manuscript presents the research results as a follow-up of the Wu et al. (2022, https://doi.org/10.1029/2022JD037406) paper, in that the model simulations were performed and analyzed using the GLM-based lightning NOx emissions. The manuscript is well written and clearly presented. However, some clarifications/corrections are needed before it can be accepted for publication. The major concern is the IC/CG ratios as described in Section 2.2 and related results/discussions in later sections. As for previous studies using lightning data from ground-based networks, such as NLDN, WWLLN, and ENTLN, the ratios are needed because the ground-based networks mainly detect CG flashes (or only the portion of CG flashes are used as in the case of NLDN). However, the GLM data provides the total lightning flashes (CG+IC) and I would assume that the detection efficiency is probably higher for IC than CG, because the CG flashes are often from the cloud base to the ground where the thick cloud layer above may reduce the detection efficiency from GLM data. Therefore, the inclusion of IC/CG ratios to calculate lightning NOx risks double counting the IC flashes. Please clarify what and how you used the IC/CG ratios when calculating LNOx. In addition, does the 0.174 Tg N of LNOx include the IC/CG ratio adjustment or it is simply produced using the total lightning flashes?

**Response:** We thank the reviewer for the thoughtful comments on the manuscript. The reviewer's concern about the use of IC/CG ratio is perhaps due to the lack of a detailed description of emissions processing in the current paper. Since a detailed description was presented by Wu et al. (2023), we only presented a short summary in section 2.2. LNO emissions used in this study were described in Wu et al. (2023) and are based on GLM-observed optical energy. As described by Wu et al. (2023), GLM-east and GLM-west observations are combined in order to reduce the impact of detection efficiency with respect to increased viewing angle, and no other adjustment is performed. The variation in per-flash NO production is merely controlled by observed optical energy. Thus, less energetic ICs produce fewer NO molecules than the more energic CGs. Then, the total emissions for each grid cell are distributed vertically according to the NASA Lightning Nitrogen Oxides Model (LNOM). LNOM uses climatology obtained from the North Alabama Lightning Mapping Array (LMA) to create LNO profiles for ICs and CGs separately. Therefore, by assuming that our total column GLM-based emission over each grid cell is for the combined ICs and CGs, the climatological IC/CG ratio from Boccippio et al. (2001) is used to vertically distribute the estimated NO. We revised the text to eliminate any ambiguity about the use of IC/CG ratio and clearly explained that the IC/CG ratio is only used for the vertical distribution of total column LNO estimates.

Furthermore, our 0.174 Tg N figure reflects the total emissions based on GLM-observed optical energy that accounts for all flashes (ICs+CGs).

Minor comments:

The last sentence of Abstract (Lines 23-34) is somewhat ambiguous and it could be implied in the prior sentence. I would suggest dropping this sentence.

**Response**: Thank you for this suggestion. The sentence has been dropped.

Lines 89-90, this desynchronization can also be resolved using lightning data assimilation with the same lightning flash data in the upstream meteorological model (https://doi.org/10.1002/2016MS000735 and https://doi.org/10.5194/gmd-15-8561-2022).

**Response**: We thank the reviewer for this useful comment. We have added these references to the manuscript.

Line 107, FOV first appears, please define it.

**Response**: FOV has been revised to "field of view".

Lines 149-150, "So even though … therefore contains uncertainty", please consider rewriting this sentence. In addition, please comments on the low detection efficiency problem for GLM over the Great Plains as shown in Wu et al. Figure 8, and Allen et al. (2020, https://doi.org/10.1029/2020JD034174).

**Response**: We thank the reviewer for this suggestion. This sentence has been rewritten for clarity. The GLM low flash DE issue is also commented on later in this paragraph.

Lines 169-172, see the Major comments.

**Response**: Section 2 is a recap of the GLM-based LNO emission model demonstrated in Wu et al. (2023). Details on the estimation of LNO emission, including how the IC-to-CG ratio was applied when producing LNO emission profiles, were discussed in Wu et al. (2023) and are not repeated here. We revised the text to eliminate any ambiguity about the use of the IC/CG ratio and clearly explained that the IC/CG ratio is only used for the vertical distribution of total column LNO estimates. The readers are also referred to Wu et al. (2023) for more details on the LNO emission model.

Lines 184-189, I believe that you performed continuous simulations during the study period, and the 5.5-day run segments were just the way the input data were organized. In essence it is just a continuous simulation, and therefore, there is no need to give the detail on how you did that (it only causes unnecessary confusions).

**Response**: Strictly speaking, the WRF model was re-initialized every 5 days and ran for 5.5 days. For the overlapping time period, i.e., the last 12 hours of the 5.5-day run segment and the first 12 hours of the next run segment, there would be some differences. We discarded the first 12 hours of each new run segment, which was replaced by the last 12 hours of the previous 5.5-day run segment. This is done to reduce the meteorological drift, while the air quality simulation is continuous. We think this detail is important, as many modelers do not consider the impact of increased forecast errors on chemistry in longer simulations. We added text to avoid confusion.

Lines 212-213, does the chemical initial and boundary condition input files contain LNOx emissions? Please clarify,

**Response**: Chemical initial and boundary conditions were extracted from the CAM-chem outputs without any modification. The CAM-chem is a component of the NCAR CESM (Emmons et al., 2020). Based on the document, their model simulation does include online calculation of LNO emission.

Section 4.1, when talking about the specific LNOx numbers and the ratios, please clarify if all the grid cells within the modeling domain or only the land-based grid cells are used and if the grid cells from Mexico and Canada are counted or not. Apparently, soil NOx and anthropogenic emissions are mainly from land (is the shipping emissions included? Figure 2 provides some hint, but specific clarification would be helpful). While counting the record numbers in Tables 5 and 6 and also in the supplementary tables, the records from all the regions don't add up to the total "Domain" records. For instance, the "Domain" records are a few hundred more than the tally from the regions for MDA8 O3 statistics, while the "Domain" records are less than the tally from the regions for NOx statistics. Please check and clarify how the "Domain" is defined.

**Response**: We thank the reviewer for such a useful comment. For Section 4.1, the analysis of emissions includes all model grid cells (US, Mexico, Canada, and the ocean). The first sentence of Section 4.1 has been revised to clarify this. Anthropogenic emissions and biogenic emissions were processed based on the EPA emission platform, which has anthropogenic emissions over Mexico and Canada, and shipping emissions. We carefully reviewed the analysis of MDA8 O3 and daily NOx, and found that the AMET-ready O3 and NOx observations include stations in Mexico. Those points have been filtered out in Tables 5 and 6 and also in the supplementary tables. In addition, for daily NOx statistics, errors were noticed for regions RM, SE, LM during July, August, and September. They have been reprocessed and the stats have been updated.

Lines 227-233, when compare the ratios with literature, please take into consideration the variation trend of anthropogenic NOx emissions over the years. In the past two decades or so, the average anthropogenic NOx emissions have been decreased by ~4-5%/year in CONUS.

**Response**: We thank the reviewer for such a useful comment. This part has been revised to address the issue. We have added statements regarding the decreasing trend of anthropogenic NOx emission over the CONUS, and pointed out that the estimates of LNO contribution given in Bond et al. (2001) should increase over the years.

Line 310, "upper troposphere that the ground level", add "at" before "the ground level".

**Response**: This has been revised.

Line 357, "As the ozone-enhanced plume", may be "As the NOx and ozone enhanced plume",

**Response**: This has been rewritten as: As the ozone enhanced plume -> As the ozone (and NOx) enhanced plume

Line 378, "in the LGNTO model run than the CNTRL" suggest changing to "in LGNTO than in CNTRL".

**Response**: This has been rewritten as: in the LGTNO model run than the CNTRL -> in the LGTNO than in the CNTRL

Lines 383-384, in addition to O3, NO can also be oxidized by O2 and OH to NO2 in the atmosphere.

**Response**: Yes, NO can also be oxidized by O2 and OH at night. But we want to emphasize the role of ozone here since the concentration of OH in the atmosphere is pretty low, and only NO oxidation by ozone can cause sharp increases in NO2 as seen in the plot (strong correlation between the drop in ozone level and NO2 increase).

Line 385, "Shortly after sunrise, due to photochemistry", I would say "due to boundary layer mixing which provides the surface precursors and photochemistry".

**Response**: The whole sentence has been revised.

The "Data Availability" and "Author Contributions" sections are missing.

**Response:** The "Data availability" and "Author Contributions" were added.

Figure 3, the color scheme is too reddish, a better color scheme is preferred (with more contrasting colors). For the difference plots, please mark (LTGNO - CNTRL) for better clarity.

**Response**: We thank the reviewer for this useful suggestion. Figure 3 has been updated with more contrasting colors. We also mark "LGTNO - CNTRL" in each difference plot.

Figure 7, suggest using difference plots (b and c) by pairing the model and Lidar observations in time and space. For example, b would be CNTRL – Lidar, and c, LTGNO – Lidar.

**Response**: We thank the reviewer for the suggestion. It would be interesting to show lidar-model difference plots here. However, since lidar curtains were missing above certain altitudes due to clouds or reduced signal-to-noise ratios at multiple timestamps, making such difference plots would look strange under these circumstances. Also, we think that making these changes won't be more helpful for the conclusions.

Figure 10, please clarify how the values are calculated on the map, is it max (LTGNO – CNTRL) for all the time steps at each grid cell?

**Response**: Yes, it is the maximum ozone difference between the LGTNO and the CNTRL for all time steps at each grid cell (within the troposphere). The caption was revised to indicate that.

In addition to O3 air quality, another often studied area is the impact of LNOx on wet/dry nitrate (NO3-) depositions. It would be nice if some comments are provided in the Introduction or Conclusions sections. Please see a recent paper in this respect (https://www.mdpi.com/2073-4433/13/8/1248).

**Response**: Thank you for this comment. Some comments have been added at the end of the conclusion section, stating the impact of LNO emission on wet and dry depositions of the aerosol nitrates (NO3-) are worth further investigation. The paper by Kang et al. (2022b) was cited as well.

---

## Author Response (AR2)

**Responses to Reviewers.**

**Response to Reviewer #1**:

Minor Comments:
L16: ozone concentration by 1-2% (or 0.3-1.5 ppbv) in the column
This statement is confusing and incorrect. Column depths should be given in molecules cm-2, concentrations in molecules cm-3, and mixing ratios in ppv or ppm. You can refer to mean mixing ratios in the column and use ppbv; however, the range 1-2% is not consistent with 0.3-1.5 pbv.
**Response**: We thank the reviewer for the comment. The numbers (1-2% and 0.3-1.5 ppbv) correspond to Figure 6, which shows "the average increase of ozone mixing ratio *in August 2019* at each model layer *within the troposphere*" rather than "in the column". We agree that using the phrase "in the column" would cause confusion. The difference in ppbv would be much larger if the stratosphere is also included. The sentence has been rewritten to avoid being misunderstood.

L21-22: Are you saying that the production rate in "your" model needs to be refined or that your study indicates that the production rate commonly used by modelers needs to be refined. I believe you mean the former, but it is not clear. Also, rather than using "refined" be specific and use "increased" or "decreased".
**Response**: We thank the reviewer for the comment. We meant "the LNO production used is our current study needs to be refined", and the value is very likely underestimated. The sentence has been revised for clarity.

L421: The range 1-2% is not consistent with 0.3-1.5 ppbv.
**Response**: We thank the reviewer for the comment. As we explained under the first minor comment: the numbers correspond to Figure 6, which shows "the average increase of ozone mixing ratio *in August 2019* at each model layer *within the troposphere*" rather than "in the column". The difference in ppbv would be much larger if the stratosphere is also included. The sentence has been revised to make it clearer.

Grammatical Comments:
L14: produces --> produced
**Response**: This was corrected.

L19: However, many of the numbers are --> These values are
**Response**: This was corrected.

L20: increases the contribution --> increases the relative contribution
**Response**: This was corrected.

L156: found relatively lower --> found to be relatively lower
**Response**: This was corrected.

L246: could be underestimated in this study --> could be underestimated in this study or overestimated in their study

**Response**: This was corrected.

L290: prediction skill --> predictive skill

**Response**: This was corrected.

L336:, which is similar to the ground-level performance as discussed in Section 4.2. --> To improve clarity, I suggest deleting this portion of the sentence

**Response**: This was corrected.

L383: can also be limited --> is also limited

**Response**: This was corrected.

L407: The elevation of air mass --> The elevation of the air mass

**Response**: This was corrected.

L422: not evenly --> unevenly

**Response**: This was corrected.

L441: needs to be increased --> could be increased

**Response**: This was corrected.

L441/442: A more reasonable LNO production rate --> A more accurate LNOx production rate

**Response**: This was corrected.

L446: particulate matters --> particulate matter

**Response**: This was corrected.